# Sustainable Scalable Mechanochemical Synthesis of CdS/Bi_2_S_3_ Nanocomposites for Efficient Hydrogen Evolution

**DOI:** 10.3390/nano14221785

**Published:** 2024-11-06

**Authors:** Zhandos Shalabayev, Abylay Abilkhan, Natalya Khan, Saparbek Tugelbay, Anar Seisembekova, Batukhan Tatykayev, Matej Balaz

**Affiliations:** 1Department of General and Inorganic Chemistry, Al-Farabi Kazakh National University, Almaty 050040, Kazakhstan; zhandos.shalabay@gmail.com (Z.S.); abylay.abilkhan@nu.edu.kz (A.A.); natalya.khan@nu.edu.kz (N.K.); saparbek.tugelbay@nu.edu.kz (S.T.); seysembekovaanar@gmail.com (A.S.); 2National Laboratory Astana, Nazarbayev University, Astana 010000, Kazakhstan; batukhan.tatykayev@nu.edu.kz; 3Institute of Geotechnics, Slovak Academy of Sciences, 04001 Kosice, Slovakia

**Keywords:** mechanochemistry, ball milling, semiconductor, cadmium sulfide, bismuth sulfide, hydrogen evolution

## Abstract

In the present study, a green, scalable, and environmentally friendly approach was developed for the fabrication of Bi_2_S_3_-decorated CdS nanoparticles with an efficient hydrogen generation ability from the water. As a sulfur source, thiourea was used. The process was completed in two stages: mechanical activation and thermal annealing. The presence of spherical CdS nanoparticles and Bi_2_S_3_ nanorods in the CdS/Bi_2_S_3_ nanocomposite was confirmed and proved by XRD, Raman spectroscopy, SEM-EDS, and TEM. The synthesized CdS/Bi_2_S_3_ nanocomposites were evaluated for their photocatalytic hydrogen evolution capabilities. The CdS/Bi_2_S_3_ photocatalyst exhibited 25% higher photocatalytic activity compared to CdS, reaching a hydrogen evolution rate of 996.68 μmol h^−1^g^−1^ (AQE 0.87%) after 3.5 h under solar-light irradiation.

## 1. Introduction

The production of hydrogen has recently garnered considerable attention due to its potential as a clean and sustainable energy source. As the global community strives to address the challenges posed by climate change and reduce reliance on fossil fuels, hydrogen emerges as a promising alternative [1,2]. Its versatility and high energy content, as well as our capacity to produce it from a range of renewable sources, underscore its significance in achieving a low-carbon future [3].

Hydrogen generation can be achieved through various methods, including steam methane reforming [4], electrolysis [5], thermochemical water splitting [6], biological processes [7,8], and photocatalytic water splitting [9]. Each method presents unique advantages and challenges, with ongoing research focused on improving efficiency, reducing costs, and minimizing environmental impacts [10]. Photocatalytic water splitting, in particular, offers a promising approach by using sunlight and semiconductor materials to directly produce hydrogen from water [11].

Semiconductor nanomaterials are promising photocatalysts for hydrogen generation due to their unique properties, such as high surface area, efficient light absorption, suitable bandgap, and catalytic activity [12]. These nanomaterials, such as TiO_2_, ZnO, g-C_3_N_4_, CdS, and their binary [13,14] and ternary composites [15,16], are designed to absorb sunlight and facilitate the separation of water molecules into hydrogen and oxygen [17]. Among them, cadmium sulfide (CdS) is of particular interest due to its optimal bandgap energy and visible-light absorption, making it a highly effective material for hydrogen production [18,19]. However, its efficiency is limited by rapid electron–hole recombination and photocorrosion. To address these issues, CdS can be decorated with metal sulfides such as ZnS, CuS, NiS, MoS_2_, Bi_2_S_3_, etc. [20]. This approach leverages synergistic effects between different semiconductors, such as improved charge-separation efficiency, expanded light-absorption range, and enhanced catalytic activity [21]. Among the above-mentioned co-catalysts, Bi_2_S_3_ attracts particular interest due to its narrow bandgap, high absorption coefficient, and excellent photocatalytic properties [22]. The incorporation of Bi_2_S_3_ into CdS nanoparticles not only improves the light absorption capacity but also facilitates better charge separation and transfer. This results in reduced recombination rates of photogenerated electron–hole pairs, significantly enhancing the photocatalytic efficiency [23,24]. Furthermore, CdS/Bi_2_S_3_ photocatalysts present notable advantages, including enhanced stability, elevated catalytic activity, adjustable band structure, and cost-effectiveness [24,25]. These attributes render them highly appealing for a spectrum of photocatalytic applications, spanning from environmental remediation to renewable energy generation [26]. However, the CdS/Bi_2_S_3_ system has notable limitations, including photocorrosion, poor stability, rapid charge recombination, and inefficient charge transfer. Other challenges involve scalability issues, limited light absorption, Bi_2_S_3_ degradation, and suboptimal photoelectrochemical performance, all hindering practical applicability [27]. Addressing the limitations of the CdS/Bi_2_S_3_ system involves surface modification for stability, heterojunction engineering and co-catalyst decoration for improved charge separation, cadmium substitution to reduce toxicity, bandgap tuning for broader absorption, optimized synthesis for scalability, and nanostructuring to enhance photoelectrochemical performance [27,28].

The literature contains several papers demonstrating that CdS/Bi_2_S_3_ can generate hydrogen through water splitting. To illustrate this point, in the work [29], Yang et al. synthesized 2D Bi_2_S_3_/CdS nanosheet arrays through a three-step complex process and utilized them as photoanodes, achieving improved photoelectrochemical hydrogen evolution. Another CdS/Bi_2_S_3_ was prepared via a one-step solvothermal method and exhibited an exceptional photocatalytic hydrogen production performance [30]. Lately, binary CdS/Bi_2_S_3_ heterostructures were fabricated using ZnO/Bi_2_S_3_ as active intermediates. The highest photocatalytic hydrogen evolution reaction rate was 3.85 mmol∙g^−1^∙h^−1^ [31].

In addition to the abovementioned approaches, CdS/Bi_2_S_3_ nanocomposites can be synthesized also through techniques such as sonochemistry [23], hydrothermal [32], wet-chemistry approach [33], ion exchange [34,35], one-pot controlled synthesis [36], in situ fabrication [37], sol–gel [38], the SILAR method [39], chemical-bath deposition [40], and mechanochemical [41,42] in order to be employed in various applications. Nevertheless, the prevailing synthesis techniques, including sol–gel, hydrothermal, and chemical-bath deposition, are encumbered by considerable limitations [43]. These include the utilization of toxic solvents, intricate reaction conditions, and scalability challenges. In contrast, mechanochemical synthesis offers a solvent-free, environmentally friendly, and scalable alternative [44,45]. By employing mechanical energy to propel the reaction, our methodology obviates the necessity for deleterious solvents, streamlines the process, and permits synthesis under ambient conditions [42]. Moreover, the ball-milling process fosters superior contact between CdS and Bi_2_S_3_, thereby enhancing charge separation and augmenting photocatalytic efficacy.

This study aims to develop a cost-effective, solvent-free, and scalable method for synthesizing Bi_2_S_3_-decorated CdS nanoparticles with enhanced photocatalytic hydrogen generation capabilities. We hypothesize that the mechanochemical approach, in conjunction with thermal annealing, will improve charge separation and minimize electron–hole recombination, thereby enhancing hydrogen-evolution performance. To the best of our knowledge, this is the first time that the mechanochemical route is applied to the solid-state production of CdS/Bi_2_S_3_ nanocomposites.

## 2. Experimental Part

### 2.1. Materials

Cd(NO_3_)_2_ 4H_2_O (99.9%, Sigma, St. Louis, MO, USA), Bi(NO_3_)_3_∙5H_2_O (97.5%, Sigma), CS(NH_2_)_2_ (99.0%, Sigma), NaOH (95.0%, Sigma), and PtCl_2_ (98.0%, Sigma) were used without further purification. Deionized water was used in all experiments.

### 2.2. Mechanochemical Synthesis of CdS Nanoparticles

The ball-milling preparation of cadmium sulfide (CdS) nanoparticles was conducted following the methodology previously outlined in [42]. In a typical mechanosynthesis, the requisite stoichiometric quantities of cadmium nitrate, thiourea, and sodium hydroxide were homogenized and transferred to the milling jar (45 mL, ZrO_2_) of the Pulverisette 7 classic line ball mill (Fritsch, Idar-Oberstein, Germany). The experimental parameters were as follows: milling duration of 10 min, rotational speed of 400 rpm, a ball-to-powder ratio of 35, and an air milling atmosphere. These parameters were found to be optimum in previous work [42] by adjusting the synthesis time, ball-to-powder ratio, and rotation speeds of the milling chamber to achieve effective size reduction while minimizing particle agglomeration and heat generation. The final product was washed three times with water and one time with ethanol.

### 2.3. Preparation of Bi_2_S_3_ NPs via Mechanical Activation with Further Annealing

Bismuth sulfide nanocrystals were prepared by combining two approaches: mechanochemical and thermal synthesis [46]. Some experimental conditions of the thermal method were modified with the aim of simplification of the selected method. First of all, the necessary amounts of Bi(NO_3_)_3_∙5H_2_O and CS(NH_2_)_2_ in a 1:6 ratio were milled for 10 min at 400 rpm, under an air atmosphere. The milling equipment was the same as that used above. After that, the resulting powder mix was annealed at 275 °C for 2 h in a muffle furnace, in an air atmosphere.

### 2.4. Fabrication of CdS/Bi_2_S_3_ Nanocomposite

The Bi_2_S_3_-decorated CdS nanocomposite was synthesized by employing a combination of the aforementioned approaches (Figure 1). Firstly, 1 g of the as-synthesized dry CdS nanoparticle powder was transferred to the 45 mL ZrO_2_ ball-mill jar. Subsequently, the precursors of bismuth sulfide (Bi(NO_3_)_3_∙5H₂O and CS(NH_2_)_2_) in a stoichiometric amount were also introduced to the milling jar. CdS nanoparticles were decorated with Bi_2_S_3_ nanoparticles at three different mass fractions: 2.5 wt.%, 5 wt.%, and 10 wt.%. Following homogenization, the mixture was milled for 10 min at 400 rpm, under an air atmosphere, and identical conditions to those as mentioned above. The resulting mixture was then transferred to a porcelain crucible and annealed at 275 °C for 2 h in a muffle furnace. The obtained final powder was used without further washing.

### 2.5. Photocatalytic Hydrogen Evolution Test

In a three-necked round bottom flask, 250 mL, a mixture was created by adding 30 mg of photocatalyst powder to a 90 mL of water, 10 mL of glycerol, and 0.3 mL of H_2_[PtCl_4_], which contains 0.3 mg. The mixture was sonicated for 5 min, and then the reactor was hermetically sealed, and a flow of pure argon Ar (99.99%) was introduced at a rate of 100 mL/min. The gas exiting the reactor was directed to a gas chromatograph (GC system 8890 Agilent with PAC analytical control, Agilent, Santa Clara, CA, USA) for online measurement of gas concentrations. In this setup, pure argon was used as the carrier gas. During the first hour of the experiment, the reaction system remained in darkness, allowing the carrier gas to purge dissolved oxygen from the aqueous alcohol solution. After one hour of argon saturation, the system was irradiated with light, causing the reduced hydrogen to be transported to the gas chromatograph alongside the argon flow. A 300 W solar-simulated lamp (Osram Vita-lux, Munich, Germany) with an intensity of 30 mW/cm^2^ was used to irradiate the mixture. Hydrogen generation was monitored every 30 min. By knowing the flow rate of the outgoing gas, the volumetric concentration of hydrogen, and the laboratory temperature and pressure, we calculated the hydrogen evaluation rate (HER). This allowed us to determine the amount of hydrogen that could be reduced on the surface of 1 g of photocatalyst per hour under these conditions. The generated hydrogen was detected every 30 min. The apparent quantum efficiency (AQE) was calculated by using the provided Equation (1) [47]:(1)$$AQE=\frac{n\times \Delta G}{W\times S\times t}\times 100\%$$
where *n* is the number of H_2_ molecules evolved in the photocatalytic process, Δ*G* is the energy needed for splitting one water molecule into H_2_ and O_2_ (237 kJ × mol^−1^), *W* is the lamp power, *S* is the irradiated area, and *t* is the reaction time.

### 2.6. Instruments and Characterization

The phase composition of the samples was examined using X-ray diffraction (XRD) on a Rigaku MiniFlex 600 X-ray diffractometer (Rigaku, Tokyo, Japan), employing copper CuKα radiation (λ = 0.15405 nm, 40 kV, 15 mA). The XRD patterns were recorded in the 2 theta region, 20–60°, using a step size of 0.1° and step time of 5 s. The Rietveld refinement of XRD data was performed using TOPAS Academic Version 6 software [48,49]. Raman spectroscopy was performed with a LabRAM HR Evolution spectrometer from Horiba Scientific (Irvine, CA, USA) utilizing a 532 nm laser light for excitation. The functional group composition and structure of the samples were assessed via Fourier-transform infrared spectroscopy (FTIR) using a Nicolet iS10 FTIR spectrometer (Thermo Fisher Scientific, Waltham, MA, USA). The absorption properties of the samples and their bandgap energy were examined using UV-Vis absorption spectroscopy, recorded with a Shimadzu 2600i spectrophotometer (Shimadzu, Kyoto, Japan). The morphology and size of the samples were analyzed using scanning electron microscopy (SEM) on a JEOL JSM-IT800 (JEOL Ltd., Tokyo, Japan) and transmission electron microscopy (TEM) on a JEOL JEM-1400 Plus. Elemental distribution within the samples was investigated using SEM X-ray energy-dispersive spectroscopy (EDS) with a ZEISS Crossbeam 540 (ZEISS, Oberkochen, Germany) equipped with an X-ray EDS detector (energy resolution approximately 130 eV). High-resolution TEM (HR-TEM) was performed on a Cs-corrected scanning transmission electron microscope (JEOL JEM-ARM200F) to determine the interplanar spacing in the crystal structure.

## 3. Results and Discussions

### 3.1. XRD Analysis

The XRD patterns of CdS, Bi_2_S_3_, and Bi_2_S_3_-decorated CdS nanoparticles (CdS/Bi_2_S_3_) are presented in Figure 2. The XRD pattern shows a mixture of the cubic hawleyite phase of CdS (PDF card no. 03-065-2887) and a hexagonal modification of CdS (greenockite, PDF card no. 01-074-9663). Regarding the peaks of the cubic phase, the peak with the greatest intensity at 2θ = 26.4° is associated with the (111) interlayer stacking, whereas the diffraction peaks with lower intensities at 43.8° and 52° are attributed to the (220) and (311) lattice planes, respectively [41,42]. According to the Rietveld refinement, the estimated content of hexagonal and cubic CdS in the sample is 56 ± 2% and 44 ± 2%. The broad nature of these diffraction peaks suggests that the CdS nanoparticles have very small dimensions, and this was also confirmed by the Rietveld refinement: the estimated crystallite size of cubic CdS was 5.2 ± 0.1 nm, and the one of hexagonal modification was 6.0 ± 0.3 nm. In the latter case, a microstrain of 3.2 ± 0.4% also contributes to peak broadening.

Figure 2 also illustrates the XRD pattern of bare Bi_2_S_3_. Apart from a few minor peaks, the observed diffraction peaks could be well-matched to an orthorhombic phase of Bi_2_S_3_ (bismuthinite, PDF card no. 01-089-8963). The estimated crystallite size was 70 ± 4 nm, so Bi_2_S_3_ nanocrystals are much larger than that of CdS. This is corroborated by a narrower character of the diffraction peaks compared to CdS. Also, a microstrain of 0.34 ± 0.02% was found to contribute to peak broadening. The intensities and positions of the identified peaks correspond well with those reported in the literature for Bi_2_S_3_ [50,51].

For the CdS/Bi_2_S_3_ sample, the XRD pattern displays prominent peaks at 26.4°, 43.8°, and 51.6°, corresponding to the (111), (220), and (311) planes of cubic CdS, respectively. The content of the hexagonal phase is higher than detected for the bare CdS sample, as confirmed by Rietveld refinement (the content of hexagonal CdS is 75 ± 2%, whereas that of the cubic phase is only 25 ± 2%). The absence of intense and sharp peaks attributable to Bi_2_S_3_ suggests a low Bi_2_S_3_ content, and it is below the detection limit of X-ray diffraction. The estimated crystallite size of the cubic CdS was even finer than in the bare CdS, namely 2.5 ± 0.04 nm. On the other hand, the crystallite size of hexagonal CdS was larger, namely 20 ± 4 nm. In this case, a microstrain of 3.3 ± 0.1% was also evidenced.

### 3.2. Raman Spectroscopy

To confirm the preparation of the cadmium sulfide and bismuth sulfide complex without structural defects, it was additionally studied with Raman spectroscopy (Figure 3a). In the CdS sample, three pronounced and broad peaks were observed, which can be attributed to the optical vibrational Raman active modes at 293 cm^−1^, 589 cm^−1^, and approximately 840 cm^−1^. As is known, these modes are attributed to the fundamental optical phonon mode (LO), the first overtone mode (2LO), and the second overtone (3LO) of CdS, respectively [52]. In the present case, the intensity of the 3LO overtone is notably low. It has been reported that the peak at 293 cm^−1^ can be shifted by grinding to a nanometer size [53,54,55]. Furthermore, alterations to the peak may result in asymmetries and a shift towards lower frequencies [56]. The observed position of the Raman scattering peak of LO microspheres has a cubic zinc blende structure. The Bi_2_S_3_ sample exhibits scattering bands at 426 and 962 cm^−1^, which are noted by the published literature’s data for crystalline Bi_2_S_3_ [57]. Other minor peaks are obscured by the background noise. As is widely acknowledged, these peaks are indicative of the optical phonon modes and surface phonon modes [58,59,60]. The final sample exhibits unambiguous evidence of CdS modes and an intense peak at 962 cm^−1^, which can be attributed to the modes of crystalline Bi_2_S_3_. It is also noteworthy that no evidence of an amorphous phase was observed in the complex. The Raman spectra demonstrate the structural integrity of the CdS/Bi_2_S_3_ composites.

### 3.3. FTIR Spectroscopy

Fourier-transform infrared (FTIR) spectroscopy was employed to investigate the vibrational characteristics and functional groups associated with the synthesized CdS/Bi_2_S_3_ composite, as well as the individual peaks of CdS and Bi_2_S_3_ samples (Figure 3b). In the case of bare CdS and CdS/Bi_2_S_3_ composite, a moderately intense absorption peak at 655 cm^−1^ was detected, corresponding to the Cd–S stretching vibration [25]. In the Bi_2_S_3_ sample, the weak peaks observed within the 500–1000 cm^−1^ range are attributed to the stretching vibration of the Bi–S bond. Similar findings have been reported by Boughdachi et al. [61]. The weak absorption peak at 875 cm^−1^ is ascribed to the out-of-plane bending vibration of the O–H bond within water molecules. The absorption bands at 1018 cm^−1^ and 1117 cm^−1^ are attributed to the C–N stretching vibration and the C=S bond of thiourea, respectively. The peak observed at 1350 cm^−1^ is attributed to the C–N stretching of tris-amine, which is also associated with C–O stretching. Furthermore, the peaks at 2051 cm^−1^ and 2161 cm^−1^ also correspond to the N=C stretch, which is indicative of isothiocyanate (–NCS). This is a consequence of the hydrolysis of thiourea that occurred during the synthesis [62]. The absorption bands observed at 1622 cm^−1^ and 3333 cm^−1^, characterized by medium intensity and broadening, were attributed to the O–H stretching vibration of water molecules, as inferred from their spectral properties [63]. These vibrations were detected due to the residual moisture within the final products that remains even after thermal annealing in the case of Bi_2_S_3_ and the CdS/Bi_2_S_3_ composite.

### 3.4. UV–Visible Spectroscopy

The ultraviolet–visible diffuse reflectance spectroscopy (UV-Vis DRS) technique was employed to characterize the light-absorption properties of pure cadmium sulfide (CdS), bismuth sulfide (Bi_2_S_3_), and CdS/Bi_2_S_3_ composite materials. As illustrated in the accompanying Figure 4a, pure CdS exhibits a visible absorption peak at 545 nm, while Bi_2_S_3_ displays the broadest absorption at 1028 nm. The CdS/Bi_2_S_3_ composite, situated in the middle of the spectrum, exhibits an absorption peak at 575 nm. In comparison to pure CdS, the CdS/Bi_2_S_3_ composite exhibits a shift from green to yellow visible light. This shift is likely to affect its visible-light absorption, leading to the formation of photogenerated hole–electron pairs under solar illumination. Tauc’s equation, (αhv)*n* = B(hv-Eg), was employed to obtain the value of the bandgap energy [64].

In this formula, α—absorption coefficients; v—frequency of light; B—edge width parameter; Eg—gap energy; and *n* is 2 for direct semiconductor and ½ for indirect semiconductor. For pure Bi_2_S_3_ and CdS, the value of *n* will be 2 for the allowed direct transition [24,30,65].

Figure 4b shows a plot of (αhv) 2 versus hv photon energy for synthesized samples that was used to calculate the bandgap energy of them. The straight line to (αhv) 2 = 0 was extrapolated, and the corresponding gaps (Eg) were found to be 1.3 eV for pure Bi_2_S_3_, 2.4 eV for CdS, and 2.31 eV for CdS/Bi_2_S_3_. The increase in the bandgap of the CdS/Bi_2_S_3_ composite compared to pure Bi_2_S_3_ and the slight reduction relative to CdS indicate an optimization of optical properties, which contributes to enhanced photocatalytic efficiency.

### 3.5. XPS Analysis

X-ray photoelectron spectroscopy (XPS) was employed to examine the elemental composition of the CdS and Bi_2_S_3_ nanoparticles, thereby facilitating the determination of the chemical states of the elements. The results demonstrated the presence of cadmium, bismuth, and sulfur on the surface of the samples. Figure 5a illustrates that the peaks at 404.48 eV and 411.38 eV correspond to the surface Cd [66], while the two intense peaks at 405.88 eV and 412.68 eV are related to the valence orbitals of Cd 3d_5_/_2_ and Cd 3d_3_/_2_ [67]. Furthermore, Figure 5b illustrates the S 2p peaks, which exhibit a weak peak at 161.97 eV and a prominent peak at 163.98 eV [67]. Figure 5c depicts the peaks for bismuth in bismuth sulfide, specifically Bi 4f, which are located at 157.98 eV and 163.08 eV. These peaks exhibit two intense peaks caused by spin–orbit splitting and correspond to Bi 4f_7_/_2_ and Bi 4f_5_/_2_, respectively. Similarly, the plot displays weak sulfur peaks at 160.88 eV and 162.08 eV, corresponding to S 2p3/2 and S 2p1/2. Notably, the peak at 160.88 eV is particularly weak and exhibits an overlap with other signals. The X-ray diffraction (XRD) and IR spectroscopy results confirm the presence of sulfur in the structure. Due to the quantum effect of spin–orbit interaction, the energy level is split into two closely spaced peaks. Therefore, in Bi_2_S_3_, sulfur has an oxidation degree of −2 (S^2−^), and the two peaks are divided at the 2p-orbitals, leading us to the conclusion that one of the peaks is overlapped. The XPS results corroborate the presence of cadmium, bismuth, and sulfur in the composite, which is to the anticipated structure of the material. Figure 5d depicts the XPS valence-band spectra, the maximum value of the tangent intersection, and the minimum value of the tangent intersection of Bi_2_S_3_ and CdS. The valence-band energy (Ev) was calculated using the following formula:(2)$$E_{V}=E_{He\;I\;exc.}-\Delta E$$

In this instance, the excitation energy of He I is represented by the symbol (He I exc.) and is equal to 21.22 eV. The binding energy of the secondary cut-off threshold is represented as Δ*E*, which is the difference between the minimum value of the tangent intersection and the width of the He I XPS spectrum. The Ev values CdS and Bi_2_S_3_ were determined to be 5.82 eV and 6.02 eV, respectively. The results obtained were used to calculate the conduction-band energy using the Eg.
(3)$$E_{C}=E_{V}-E_{g}$$

In this context, the term Ev refers to the energy associated with the valence band, and Eg denotes the bandgap energy. The Ec values were determined to be 4.52 eV and 3.62 eV. The following methods were employed to ascertain the energy levels of Bi_2_S_3_ and CdS about the NHE (normal hydrogen electrode) potential. In this context, Ev is used to denote the upper boundary of the valence band (VB), while Ec is used to denote the lower boundary of the conduction band (CB). To ascertain the potential of the valence band relative to the NHE, the value of Ev was subtracted from the Fermi energy (4.44 eV).
(4)$$VB=E_{V}-4.44\;eV$$

The VB values CdS and Bi_2_S_3_ were determined to be 1.38 eV and 1.58 eV, respectively. To ascertain the conduction band potential of the NHE standard, the Ec value was subtracted from the Fermi energy, which was determined to be 4.44 eV.
(5)$$CB=E_{C}-4.44\;eV$$

The CB values CdS and Bi_2_S_3_ were determined to be 0.08 eV and −0.82 eV, respectively.

The XRD analysis (Figure 2) revealed the presence of two distinct crystal structures of CdS in our samples, namely the cubic (hawleyite) and hexagonal (greenockite) phases. These polymorphs display disparate electronic properties, notably in their conduction-band minimum (CBM) and valence-band maximum (VBM). The hexagonal CdS phase typically exhibits a higher CBM and a more negative VBM in comparison to the cubic phase [68]. Although these discrepancies are relatively minor, typically within a few tenths of an electronvolt, they can impact photocatalytic behavior, particularly with regard to bandgap and charge-separation efficiency [69].

In the context of the CdS/Bi_2_S_3_ composite, both CdS phases contribute to the overall photocatalytic activity. The coupling with Bi_2_S_3_ is anticipated to facilitate charge transfer, thereby mitigating any potential adverse effects resulting from the differences in band positions between the two CdS polymorphs. Furthermore, the enhanced stability of hexagonal CdS, as indicated by the Rietveld refinement results, suggests its favorable role in forming heterojunctions, which can improve charge-separation efficiency and, consequently, the overall photocatalytic performance of the composite system [70,71].

### 3.6. SEM Analysis

The SEM micrographs for all three samples are provided in Figure 6. The particles of CdS nanocrystals are very fine. In Figure 6b, larger agglomerates in micrometer-size are decorated with numerous fine grains with a size lower than 100 nm. The particles are of an undefined shape, typical for the mechanochemical synthesis [72]. The morphology of Bi_2_S_3_ is lumpy; however, in the center of Figure 6d, rod-like structures could be evidenced. The diameter of these rods seems to be around 200 nm, while their length exceeds one micron. The nano-rod structures of Bi_2_S_3_ are well-known [73]. The morphology of the composite resembles that of CdS; however, the particles are more compact, and the level of agglomeration seems to be larger than in the case of pure CdS (Figure 6f). It is possible that the lumpy-like Bi_2_S_3_ served the role as a glue, bringing CdS particles into closer contact.

To investigate the distribution of the present elements, EDS elemental mapping was performed. The results show the identical distribution of Bi, Cd, and S elements (Appendix A), thus confirming the good homogeneity of the sample and efficient intermixing between CdS and Bi_2_S_3_ particles.

### 3.7. TEM Analysis

The three products were also subjected to TEM analysis (Figure 7). The observations from the SEM analysis were confirmed, namely the fine character of the CdS sample (Figure 7a), rod and lumpy-like structure of Bi_2_S_3_ (Figure 7b), and larger agglomerates for the composite (Figure 7c,d). The individual nanoparticles of CdS in Figure 7a seem to be of spherical shape, and their size appears to be within a few nanometers. The rod visible in Figure 7b is thinner than the one present in the SEM image (Figure 6d), this time being around 30 nm thick. The length is consistent with the SEM observations mentioned above. Regarding the composite, the agglomerate of few hundreds of nanometers can be seen in Figure 7c, while in Figure 7d, with lower magnification, among the majority of the spherical CdS nanoparticles, a rod, most probably belonging to Bi_2_S_3_, can be found, this is zoomed in in the inset. The HR-TEM images in Figure 7e,f confirm the very fine character of the CdS nanoparticles, being smaller than 10 nm. The interplanar distance of 0.34 nm is consistent with the (111) crystallographic plane of the CdS phase (Figure 7f).

### 3.8. Photocatalytic Hydrogen Evolution Results

Before initiating light exposure, control experiments were carried out in the dark. These control experiments indicated that hydrogen (H_2_) is not generated without light. Figure 8a illustrates the photocatalytic hydrogen evolution rate (HER), showing a gradual increase in H_2_ production upon the start of light irradiation. The synthesized materials were exposed to 5 h of solar-light irradiation (30 mW/cm^2^). The photocatalytic activity of nanocomposites containing 2.5, 5, and 10% Bi_2_S_3_ was tested and compared. The results are presented in Figure 8a. The highest photocatalytic activity was shown by the nanocomposite containing 5% Bi_2_S_3_. The highest HER was observed in the CdS/Bi_2_S_3_(5)/Pt sample, reaching 996.68 μmol h^−1^g^−1^ at 3.5 h of irradiation and remaining stable for the rest of the process. In contrast, the CdS/Pt sample reached its peak HER at the 4th hour, with a value of 748.42 μmol h^−1^g^−1^, but its activity slightly decreased by the 5th hour. The Bi_2_S_3_/Pt and pure CdS/Bi_2_S_3_ samples did not produce any H_2_.

As shown in Figure 8b, the apparent quantum efficiency (AQE) of the CdS/Bi_2_S_3_(5)/Pt at 3.5 h of solar-light irradiation was 0.87%, while for CdS/Pt, at its peak HER at the 4th hour, the AQE was 0.67%. The AQE values at 30-minute intervals for CdS/Bi_2_S_3_/Pt and CdS/Pt are detailed in Figure 8b.

These results clearly show that bare CdS has lower activity, and Bi_2_S_3_ is inactive even with a Pt co-catalyst. The combination of the two metal sulfides with the Pt co-catalyst resulted in a synergistic effect, increasing H_2_ evolution by 1.3 times, if compared to CdS/Bi_2_S_3_(5)/Pt and CdS/Pt.

Table 1 offers a detailed comparative review of the photocatalytic hydrogen-generation rates observed in the present study, in comparison with those reported in the literature for several CdS, Bi_2_S_3_ nanoparticle, and CdS/Bi_2_S_3_ nanocomposite synthesis methods. The collected data suggest that the photocatalytic hydrogen-evolution capability of the sample obtained in the present study is on par with that achieved through alternative methods. Moreover, the eco-friendly nature of this approach, which avoids the use of hazardous organic solvents, positions it as a more favorable option compared to some of the methods presented in the table.

The development of photocatalytic hydrogen evolution has led to the creation of a number of highly effective semiconductor materials, which have the capacity to markedly enhance the efficiency of hydrogen production. Among these, the Cu/CdS/MnO_x_ heterostructure is particularly noteworthy, exhibiting an exceptional hydrogen evolution rate (HER) of 5965.03 μmol h^−^¹g^−^¹. This is largely attributed to its effective charge-separation mechanisms, which optimize the utilization of photogenerated carriers [74]. Similarly, a ZnIn_2_S_4_ photocatalyst integrated with single-atom catalysts has demonstrated an exceptional performance, achieving a HER of up to 12,416.8 μmol h^−1^g^−1^ and an AQE of 4.9%. This performance highlights the significant advantages of single-atom activation in enhancing photocatalytic activity [75]. Furthermore, the W_18_O_49_/S heterojunction has been documented to demonstrate a robust performance, with an HER of 5373 μmol h^−1^g^−1^, which serves to exemplify the advancements in photocatalytic systems [74].

In comparison, the CdS/Bi_2_S_3_/Pt composite displays a HER of 996.68 μmol h^−1^g^−1^ and an AQE of 0.87%. Although these findings do not rank our composite among the most effective photocatalysts, they nevertheless demonstrate its potential for further enhancement. The current performance indicates that, with targeted improvements and innovative modifications, this composite could be developed into a more efficient photocatalyst in future research endeavors [76].

To comprehensively evaluate the photocatalytic performance of the synthesized nanocomposite, a series of systematic experiments were conducted to measure its efficiency in photocatalytic hydrogen evolution, employing various sacrificial agents. Alongside the use of glycerol, which is a commonly used organic sacrificial agent, additional tests were carried out using inorganic salts, namely sodium sulfide and sodium sulfite (in a 1 M concentration), as well as methanol (utilized at a 10% volume fraction). These experiments aimed to determine the influence of different sacrificial agents on the rate of hydrogen production and assess the versatility of the nanocomposite under varying chemical environments.

The experimental results, as depicted in Figure 9a, unequivocally demonstrate that the nanocomposite exhibits a consistently high photocatalytic activity across the different sacrificial agents tested. Notably, the hydrogen evolution rate when using sodium sulfide/sodium sulfite reached 980 μmol h^−1^g^−1^, closely mirroring the performance observed with glycerol, which suggests that the nanocomposite maintains its photocatalytic efficiency in both organic and inorganic environments. Furthermore, the use of methanol as a sacrificial agent yielded a slightly reduced hydrogen production rate of 870 μmol h^−1^g^−1^, indicating a modest decrease in activity compared to inorganic salts, yet still reflecting significant catalytic efficacy.

In addition to evaluating the photocatalytic activity, the long-term stability of the photocatalyst was assessed through five consecutive cycles of hydrogen generation, with glycerol employed as the sacrificial agent. As illustrated in Figure 9b, the hydrogen yield after the fifth cycle remained at 890 μmol h^−1^g^−1^, which corresponds to 89% of the hydrogen output recorded during the initial cycle. This minimal decrease in performance highlights the excellent durability of the photocatalyst under repeated use. The ability to maintain such a high hydrogen production rate across multiple cycles demonstrates that the photocatalyst not only performs efficiently but also exhibits high structural and functional stability.

**Table 1 nanomaterials-14-01785-t001:** Hydrogen-production ability of CdS, Bi_2_S_3_, and CdS/Bi_2_S_3_ nanocomposites.

Synthetic Method	Catalyst	Co-Catalyst	Precursors	Experimental Conditions		Light Source	HER, mmol g^−1^h^−1^	Ref.
				Synthesis Time	Temperature, t °C			
Biomolecule-assisted	CdS	-	Cd(NO_3_)_2_∙4H_2_O, glutathione (GSH)	24 h	120	Simulated sunlight	0.085	[77]
Hydrothermal	CdS	-	CdCl_2_, Na_3_PO_4_·12H_2_O, Na_2_S∙9H_2_O, H_2_PtCl_6_·6H_2_O	12 h	180	Simulated sunlight	0.0051	[78]
	CdS	Pt					0.122	
Mechanochemical	CdS	NiS	Cd(NO_3_)_2_∙4H_2_O, CS(NH_2_)_2_, NaOH, NaCl, Ni(NO_3_)_2_, Na_2_S∙9H_2_O	10 min	RT	Simulated sunlight	0.192	[42]
Hydrothermal	Bi_2_S_3_	-	Bi(NO_3_)_3_∙5H_2_O, CO(NH_2_)_2_, Na_2_S∙9H_2_O	12 h	120	Simulated sunlight	0.271	[79]
Hydrothermal	CdS/Bi_2_S_3_	-	Bi(NO_3_)_3_∙5H_2_O, Cd(NO_3_)_2_∙4H_2_O, CS(NH_2_)_2_	10 h	150	Simulated sunlight	0.38	[30]
	CdSQDs/Bi_2_S_3_	-	Cd(NO_3_)_2_∙4H_2_O, EG, DMF, C_2_H_4_NS, Bi(NO_3_)_3_∙5H_2_O	9 h	200	Simulated sunlight	41.67	[80]
Intermediate-assisted chemical route	CdS/Bi_2_S_3_	-	Bi(NO_3_)_3_∙5H_2_O, KOH, CS(NH_2_)_2_, CH_3_OH, Zn(CH_3_COO)_2_⋅2H_2_O, Cd(CH_3_COO)_2_·2H_2_O	4 h 10 min	200	Simulated sunlight	3.85	[31]
Sonochemical	Bi_2_S_3_/CdS	-	Cd(CH_3_COO)_2_∙2H_2_O, Bi(NO_3_)_3_∙5H_2_O, CH_3_CSNH_2_	1 h	60	Visible light	5.5	[23]
Mechanochemical	CdS/Bi_2_S_3_	Pt	Cd(NO_3_)_2_∙4H_2_O, PtCl_2_, Bi(NO_3_)_3_∙5H_2_O, CS(NH_2_)_2_	2 h 10 min	275	Simulated sunlight	0.997	this work

#### The Mechanism of Photocatalytic Hydrogen Evolution

The positions of the conduction band (CB) and valence band (VB) in both CdS and Bi_2_S_3_ semiconductors can be determined using the provided formula:(6)$$E_{CB}=X-E_{e}-{0.5E}_{g}$$
(7)$$E_{VB}=X-E_{e}+{0.5E}_{g}$$

In this context, *E_CB_* and *E_VB_* represent the conduction-band and valence-band potential, respectively. *X* denotes the semiconductor’s electronegativity, *E_e_* is the energy of free electrons on the hydrogen scale (approximately 4.5 eV), and *E_g_* is the estimated bandgap value of the prepared sample, derived from UV–Vis diffuse reflectance spectrum results. The X values for CdS and Bi_2_S_3_ are 5.04 eV and 5.27 eV, respectively [81,82]. According to our calculation (Equations (6)–(9)) on XPS valence band spectra, the maximum value of the tangent intersection, and the minimum value of the tangent intersection of CdS and Bi_2_S_3_ samples, the conduction band potentials for CdS and Bi_2_S_3_ are calculated to be −0.82 eV and 0.08 eV, while their valence band potentials are approximately +1.58 eV and +1.38 eV, respectively.

In the energy-band structures of components in the CdS/Bi_2_S_3_ photocatalyst and the migration pathways of photogenerated carriers, an inherent electric field forms between CdS and Bi_2_S_3_. This field establishes a balanced Fermi level (*E_F_*), effectively suppressing charge carrier recombination and enabling photocatalytic H_2_ evolution. Numerous studies have detailed S-scheme heterojunctions composed of reduction photocatalysts with higher E_F_ and lower work functions, and oxidative photocatalysts with lower E_F_ and higher work functions, leveraging their band potentials synergistically [83,84,85]. Recent studies indicate the feasibility of constructing an S-scheme CdS/Bi_2_S_3_ heterojunction [25,31]. Initially, CdS exhibits a smaller work function and higher E_F_ compared to Bi_2_S_3_ [24]. The conduction band energy of bismuth sulfide (Bi_2_S_3_) is below zero in the energy diagram, making hydrogen reduction on the surface of bismuth sulfide nanoparticles, when used as a catalyst, unfeasible. This conclusion is supported by the absence of detectable hydrogen in photocatalytic experiments involving bismuth sulfide with platinum deposition (Bi_2_S_3_/Pt); see Figure 10. As a result, the dominant mechanism for hydrogen generation is attributed to the S-scheme. In this mechanism, during photocatalytic hydrogen generation using the (CdS/Bi_2_S_3_)Pt composite, hydrogen reduction primarily occurs at the surface of cadmium sulfide (CdS) nanoparticles. This facilitates the efficient transfer of excited electrons from CdS to platinum, enabling hydrogen reduction to proceed effectively at the Pt sites.

Upon interfacial contact, electrons migrate from CdS to Bi_2_S_3_ until reaching an equilibrium, *E_F_* (Figure 10), resulting in CdS becoming positively charged and Bi_2_S_3_ becoming negatively charged, establishing a hole-rich and electron-rich interface in the space charge region, generating an internal electric field (IEF). Under light irradiation, this band bending drives the interfacial recombination of photogenerated electrons and holes [86]. Photogenerated electrons on the CV of Bi_2_S_3_ and holes on the VB of CdS are segregated by band bending and Coulomb attraction, preserving their superior redox capabilities. Electrons enriched on the CB of CdS with a more negative potential (−0.73 eV) transferred to Pt nanoparticles, as per the S-scheme model, swiftly reduce H^+^, which was adsorbed on Pt surface, to generate *H*_2_ in the presence of sacrificial agents:(8)$$CdS/{Bi}_{2}S_{3}+h\nu \;\to CdS\;\left(e_{CB}^{-}+h_{VB}^{+}\right)/{Bi}_{2}S_{3}\;\left(e_{CB}^{-}+h_{VB}^{+}\right)$$
(9)$$CdS\;\left(e_{CB}^{-}+h_{VB}^{+}\right)/{Bi}_{2}S_{3}\;\left(e_{CB}^{-}+h_{VB}^{+}\right)\;\to \;CdS\;\left(e_{CB}^{-}\right)+{Bi}_{2}S_{3}\left(h_{VB}^{+}\right)$$
(10)$$h^{+}({Bi}_{2}S_{3,VB})+C_{3}H_{8}O_{3}^{-}\to {CO}_{2}+H_{2}O$$
(11)$${2H}^{+}+{2e}^{-}\;({CdS}_{CB})\to \;H_{2}$$

## 4. Conclusions

In situ solid-state synthesis of CdS/Bi_2_S_3_ nanocomposites via the ball-milling technique, with further thermal annealing, was successfully performed in this study. The morphology, structure, and composition of the individual CdS and Bi_2_S_3_ samples, as well as the composite CdS/Bi_2_S_3_, were thoroughly characterized using XRD, Raman spectroscopy, FTIR, SEM, EDS, and TEM techniques. The XRD analysis and Rietveld refinement revealed that both the CdS and composite samples contain cubic and hexagonal phases of CdS. The bandgap energies were found to be 2.4 eV for CdS, 1.3 eV for Bi_2_S_3_, and 2.31 eV for the CdS/Bi_2_S_3_ composite. Moreover, synthesized samples were tested for their ability to perform photocatalytic hydrogen evolution. The highest HER was observed in the CdS/Bi_2_S_3_/Pt sample, reaching 996.68 μmol h^−1^g^−1^ (AQE 0.87%) at 3.5 h of solar-light irradiation. In contrast, the CdS/Pt sample reached its peak HER at the 4th hour, with a value of 748.42 μmol h^−1^g^−1^ (AQE 0.67%). The Bi_2_S_3_/Pt sample did not produce any H_2_. Thus, the obtained results showed that combining CdS, Bi_2_S_3_, and a Pt co-catalyst produced a synergistic effect, increasing the HER by 1.3 times when comparing CdS/Bi_2_S_3_/Pt to CdS/Pt. The nanocomposite exhibits a strong photocatalytic hydrogen-generation performance with various sacrificial agents and retains significant stability over repeated use. These findings highlight the material’s potential for sustainable hydrogen production in energy applications.

## Figures and Tables

**Figure 1 nanomaterials-14-01785-f001:**
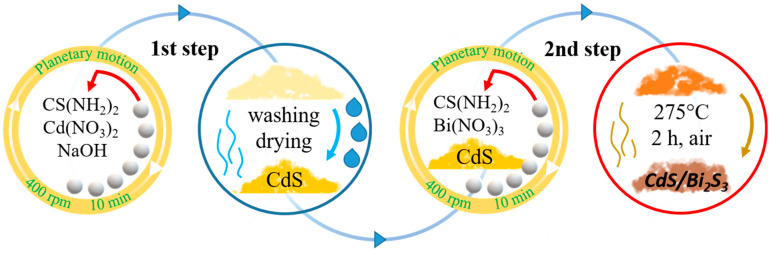
Illustration scheme of CdS/Bi_2_S_3_ nanocomposite’s preparation.

**Figure 2 nanomaterials-14-01785-f002:**
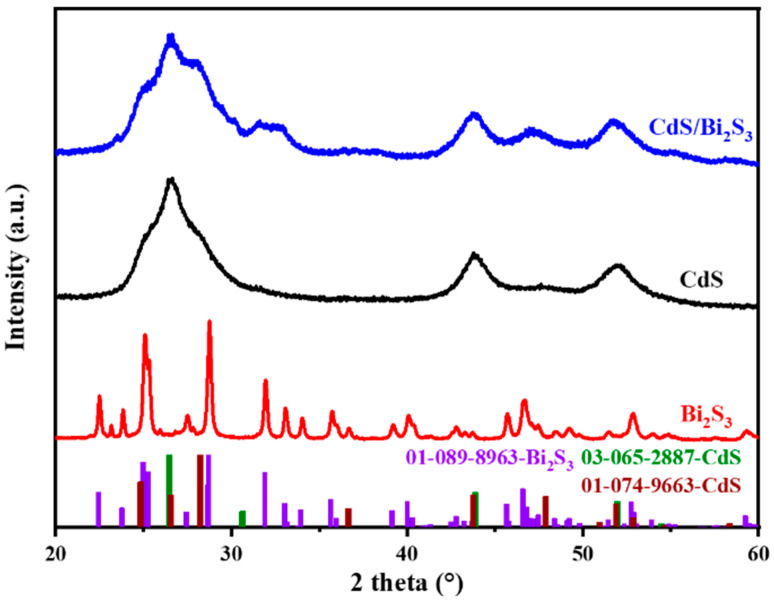
XRD patterns of the CdS, Bi_2_S_3_, and CdS/Bi_2_S_3_ samples.

**Figure 3 nanomaterials-14-01785-f003:**
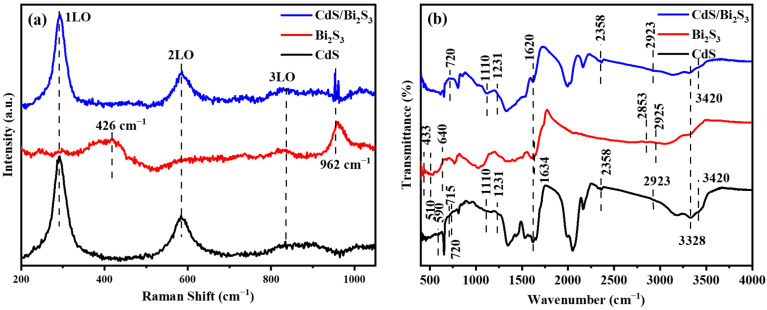
Raman spectra (**a**) and FTIR spectra (**b**) of the bare CdS, Bi_2_S_3_, and CdS/Bi_2_S_3_ nanocomposite.

**Figure 4 nanomaterials-14-01785-f004:**
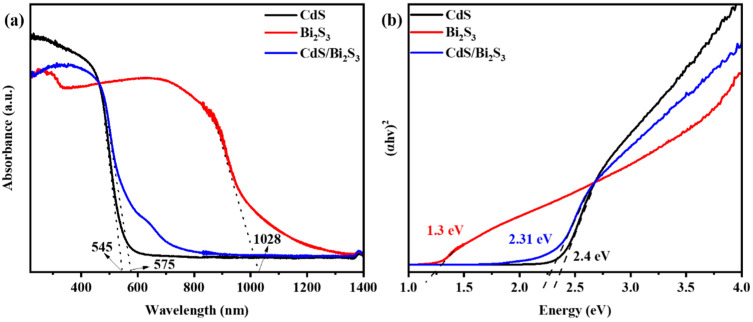
UV-Vis spectra (**a**) and Tauc plot graph (**b**) of as-synthesized CdS, Bi_2_S_3_, and CdS/Bi_2_S_3_ samples.

**Figure 5 nanomaterials-14-01785-f005:**
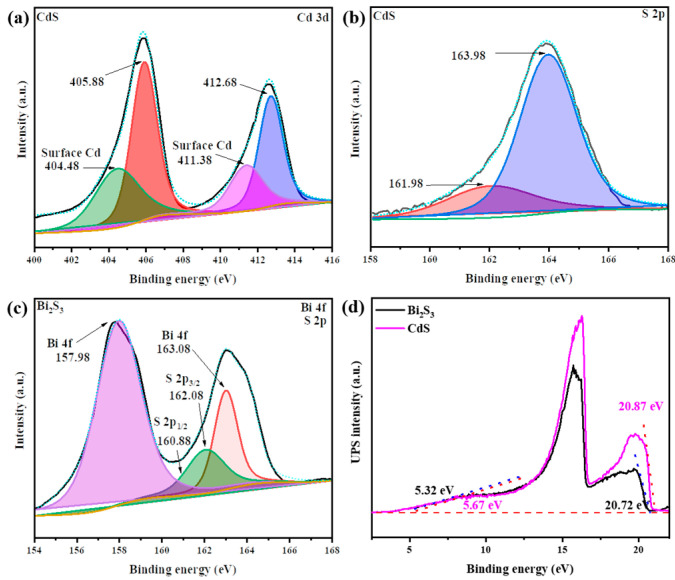
XPS spectra of the CdS (**a**,**b**) and Bi_2_S_3_ (**c**) samples. (**d**) XPS valence-band spectra, the maximum value of the tangent intersection and the minimum value of the tangent intersection of CdS and Bi_2_S_3_ samples.

**Figure 6 nanomaterials-14-01785-f006:**
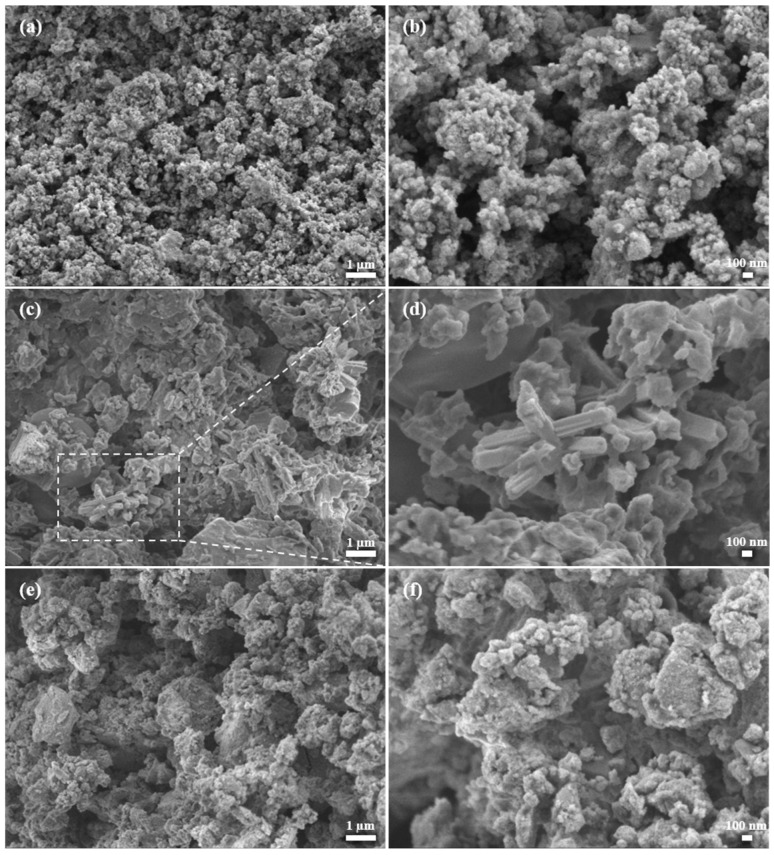
SEM images of the CdS (**a**,**b**), Bi_2_S_3_ (**c**,**d**), and CdS/Bi_2_S_3_ (**e**,**f**) samples.

**Figure 7 nanomaterials-14-01785-f007:**
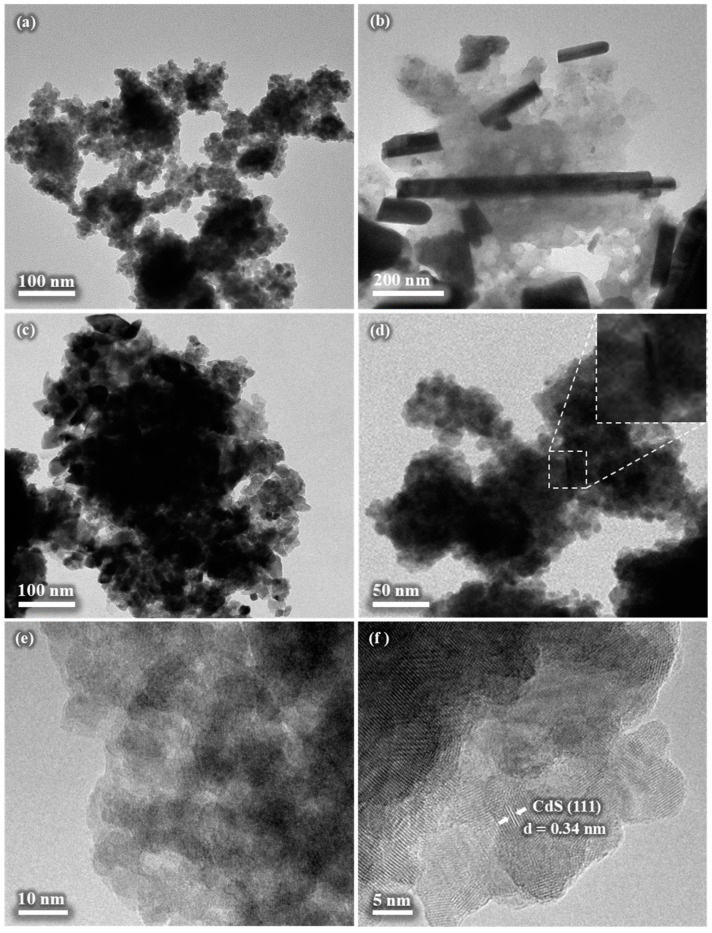
TEM images of the CdS (**a**), Bi_2_S_3_ (**b**), CdS/Bi_2_S_3_ (**c**,**d**), and HR-TEM images of CdS/Bi_2_S_3_ (**e**,**f**) samples.

**Figure 8 nanomaterials-14-01785-f008:**
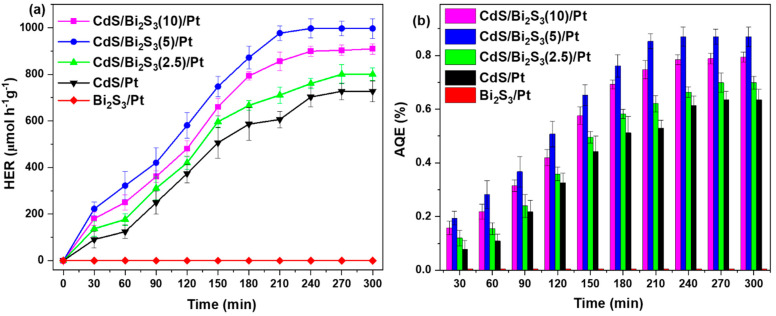
HER (**a**) and AQE values (**b**) at 30-minute intervals of the CdS, Bi_2_S_3_, and CdS/Bi_2_S_3_ samples.

**Figure 9 nanomaterials-14-01785-f009:**
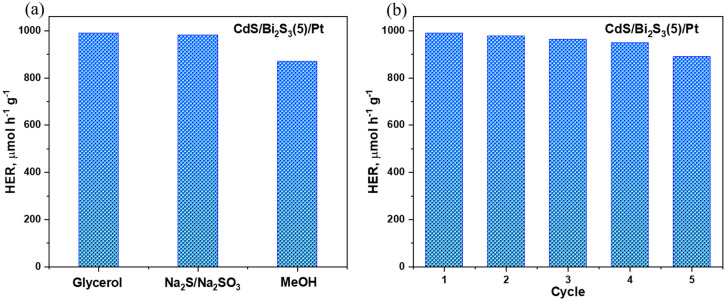
Study of photocatalytic activity of nanocomposite (**a**) using different sacrificial agents and (**b**) cyclic experiments to study the stability of the photocatalyst.

**Figure 10 nanomaterials-14-01785-f010:**
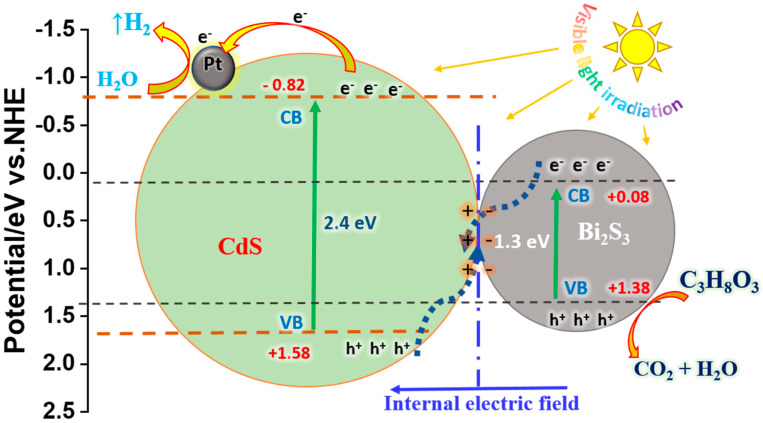
Energy band-structure diagram of CdS/Bi_2_S_3_ and S-scheme pathway for the photocatalytic mechanism of photocatalytic H_2_ evolution.

## Data Availability

The original contributions presented in the study are included in the article/Appendix A, further inquiries can be directed to the corresponding author.

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
