# Peer review of "Sustainable Scalable Mechanochemical Synthesis of CdS/Bi2S3 Nanocomposites for Efficient Hydrogen Evolution"

_nanomaterials, 2024, doi:10.3390/nano14221785_

Round 1
Reviewer 1 Report
Comments and Suggestions for Authors
The manuscript develops a green preparation method to realize the synthesis of Bi2S3/CdS composites with high hydrogen evolution capability. Thermal annealing and mechanochemical ball milling were used for the first time in solid state production of Bi2S3/CdS nanocomposites. The preparation scheme is novel and saves energy. After supplementing relevant key tests, the manuscript may be considered for acceptance.
1. Various nanomaterials have been developed for photocatalytic hydrogen evolution. What are the advantages of Bi2S3/CdS? Please add some discussions by citing typical references, such as Coordination Chemistry Reviews 2024, 502, 215612; Trends In Chemistry 2022, 4 (9), 792-806.
2. Does Bi2S3/CdS have photocatalytic activity without Pt?
3. According to the bandgap position relationship of Bi2S3/CdS, the heterojunction type constructed by the author should be biased towards type I rather than type S. Provide ESR and XPS data to prove the construction of S-scheme heterojunction.
4. Cycle stability test of hydrogen evolution is suggested to be added.
5. XRD shows that CdS has two crystal types. Is the relationship between conduction band and valence band consistent between these two crystal types?
6. Bi2S3, CdS and their composites have been studied a lot in the literature on hydrogen evolution, please add some comparisons on the performance of this work and other researches.
7. Does the relative ratio of Bi2S3 and CdS have effect on hydrogen evolution performance?
8. Comparison of hydrogen evolution capacity of photocatalyst with different sacrificial agents.
Comments on the Quality of English LanguageMinor editing of English language is required.
Author Response
Comments 1: Various nanomaterials have been developed for photocatalytic hydrogen evolution. What are the advantages of Bi2S3/CdS? Please add some discussions by citing typical references, such as Coordination Chemistry Reviews 2024, 502, 215612; Trends In Chemistry 2022, 4 (9), 792-806. |
Response 1: Thank you for pointing this out. We agree with this comment. Therefore, we have pointed other main advantages of CdS/Bi2S3 photocatalysts and cited relevant research papers including one of suggested a reviewer. The following text has been added to the text of the manuscript: “Furthermore, CdS/Bi₂S₃ photocatalysts present notable advantages, including enhanced stability, elevated catalytic activity, adjustable band structure, and cost-effectiveness [20,21]. These attributes render them highly appealing for a spectrum of photocatalytic applications, spanning environmental remediation to renewable energy generation [22]”. |
Comments 2: Does Bi2S3/CdS have photocatalytic activity without Pt? |
Response 2: Thank you for pointing this out. Experiments on photocatalytic hydrogen generation without platinum co-catalyst have been carried out, but pure CdS/Bi2S3 composite without co-catalyst did not produce any hydrogen. |
Comments 3: According to the bandgap position relationship of Bi2S3/CdS, the heterojunction type constructed by the author should be biased towards type I rather than type S. Provide ESR and XPS data to prove the construction of S-scheme heterojunction. |
Response 3: Thank you for your comment! Due to time constraints at our university laboratory, we were unable to perform ESR analysis. However, we conducted XPS analyses on the CdS and Bi2S3 samples, including XPS valence band spectra, the maximum value of the tangent intersection, and the minimum value of the tangent intersection for both samples. The results are presented in Figure 5, and the corresponding discussion of these findings is provided above the figure. These results also influenced the construction of the energy diagram, Figure 9 was edited. In the mechanism discussion, we included additional text where we indirectly demonstrate the dominance of the S scheme in the hydrogen generation process. The following text has been added to the text of the manuscript: “The conduction band energy of bismuth sulfide (Bi2S3) is below zero in the energy diagram, making hydrogen reduction on the surface of bismuth sulfide nanoparticles, when used as a catalyst, unfeasible. This conclusion is supported by the absence of detectable hydrogen in photocatalytic experiments involving bismuth sulfide with platinum deposition (Bi2S3/Pt). As a result, the dominant mechanism for hydrogen generation is attributed to the S-scheme. In this mechanism, during photocatalytic hydrogen generation using the (CdS/Bi2S3/Pt) composite, hydrogen reduction primarily occurs at the surface of cadmium sulfide (CdS) nanoparticles. This facilitates the efficient transfer of excited electrons from CdS to platinum, enabling hydrogen reduction to proceed effectively at the Pt sites”. |
Comments 4: Cycle stability test of hydrogen evolution is suggested to be added. |
Response 4: Thank you for pointing this out. We agree with this comment. Therefore, we performed a long-term stability test of the photocatalyst through five consecutive cycles of hydrogen generation. The following text has been added to the text of the manuscript: “In addition to evaluating the photocatalytic activity, the long-term stability of the photocatalyst was assessed through five consecutive cycles of hydrogen generation, with glycerol employed as the sacrificial agent. As illustrated in Figure 8b, the hydrogen yield after the fifth cycle remained at 890 μmol h⁻¹g⁻¹, which corresponds to 89% of the hydrogen output recorded during the initial cycle. This minimal decrease in performance highlights the excellent durability of the photocatalyst under repeated use. The ability to maintain such a high hydrogen production rate across multiple cycles demonstrates that the photocatalyst not only performs efficiently but also exhibits high structural and functional stability”. |
Comments 5: XRD shows that CdS has two crystal types. Is the relationship between conduction band and valence band consistent between these two crystal types? |
Response 5: We thank the reviewer for valuable comments. We agree with this comment. Thus, the implications of this mixed phase composition on photocatalytic activity was discussed. The following text has been added to the manuscript: “The XRD analysis (Figure 2) revealed the presence of two distinct crystal structures of CdS in our samples, namely the cubic (hawleyite) and hexagonal (greenockite) phases. These polymorphs display disparate electronic properties, notably in their conduction band minimum (CBM) and valence band maximum (VBM). The hexagonal CdS phase typically exhibits a higher CBM and a more negative VBM in comparison to the cubic phase [60]. Although these discrepancies are relatively minor, typically within a few tenths of an electronvolt, they can impact photocatalytic behavior, particularly with regard to band gap and charge separation efficiency [61]. In the context of the CdS/Bi₂S₃ composite, both CdS phases contribute to the overall photocatalytic activity. The coupling with Bi₂S₃ is anticipated to facilitate charge transfer, thereby mitigating any potential adverse effects resulting from the differences in band positions between the two CdS polymorphs. Furthermore, the enhanced stability of hexagonal CdS, as indicated by the Rietveld refinement results, suggests its favorable role in forming heterojunctions, which can improve charge separation efficiency and, consequently, the overall photocatalytic performance of the composite system [62,63]”. |
Comments 6: Bi2S3, CdS and their composites have been studied a lot in the literature on hydrogen evolution, please add some comparisons on the performance of this work and other researches. |
Response 6: Thank you for pointing this out. We agree with this comment. Therefore, we performed a literature review on the photocatalytic hydrogen ability of all CdS/Bi2S3 nanocomposites and compared their experimental results with others in the review-table (table 1). Additionally, following text has been added to the text of the manuscript: “Table 1 offers a detailed comparative review of the photocatalytic hydrogen generation rates observed in the present study, in comparison with those reported in the literature for several CdS, Bi2S3 nanoparticles, and CdS/Bi2S3 nanocomposite synthesis methods. The collected data suggests that the photocatalytic hydrogen evolution capability of the sample obtained in the present study is on par with that achieved through alternative methods. Moreover, the eco-friendly nature of this approach, which avoids the use of hazardous organic solvents, positions it as a more favorable option compared to some of the methods presented in the table”. |
Comments 7: Does the relative ratio of Bi2S3 and CdS have effect on hydrogen evolution performance? |
Response 7: Additional photocatalytic hydrogen degradation experiments were carried out using nanocomposites with different bismuth sulfide contents, where the content was 2.5, 5 and 10%. The following text has been added to the text of the manuscript: «The photocatalytic activity of nanocomposites containing 2.5, 5 and 10% Bi2S3 were tested and compared. The results are presented in Figure 7a. The highest photocatalytic activity was shown by the nanocomposite containing 5% Bi2S3.». Figure 7 has been changed. |
Comments 8: Comparison of hydrogen evolution capacity of photocatalyst with different sacrificial agents. |
Response 8: Thank you for pointing this out. We agree with this comment. Therefore, we performed an additional photocatalytic measurement experiments. a series of experiments were conducted to measure photocatalytic hydrogen generation using different sacrificial agents. The following text has been added to the text of the manuscript: “To comprehensively evaluate the photocatalytic performance of the synthesized nanocomposite, a series of systematic experiments were conducted to measure its efficiency in photocatalytic hydrogen evolution, employing various sacrificial agents. Alongside the use of glycerol, which is a commonly used organic sacrificial agent, additional tests were carried out using inorganic salts, namely sodium sulfide and sodium sulfite (in a 1 M concentration), as well as methanol (utilized at a 10% volume fraction). These experiments aimed to determine the influence of different sacrificial agents on the rate of hydrogen production and assess the versatility of the nanocomposite under varying chemical environments. The experimental results, as depicted in Figure 9a, unequivocally demonstrate that the nanocomposite exhibits a consistently high photocatalytic activity across the different sacrificial agents tested. Notably, the hydrogen evolution rate when using sodium sulfide/sodium sulfite reached 980 μmol h⁻¹g⁻¹, closely mirroring the performance observed with glycerol, which suggests that the nanocomposite maintains its photocatalytic efficiency in both organic and inorganic environments. Furthermore, the use of methanol as a sacrificial agent yielded a slightly reduced hydrogen production rate of 870 μmol h⁻¹g⁻¹, indicating a modest decrease in activity compared to inorganic salts, yet still reflecting significant catalytic efficacy”. |

Reviewer 2 Report
Comments and Suggestions for Authors
This manuscript presents a novel approach for the green synthesis of CdS/Bi2S3 nanocomposites using mechanochemical methods, with a focus on their application in photocatalytic hydrogen evolution. The authors have conducted a comprehensive study, employing various characterization techniques to analyze the structural, morphological, and optical properties of the synthesized materials. The work demonstrates a good understanding of the underlying mechanisms and provides valuable insights into the synergistic effects of combining CdS and Bi2S3 for enhanced photocatalytic performance. While the study is well-executed and presents interesting findings, there are several areas where improvements could be made to enhance the overall quality and impact of the manuscript.
1.The discussion of mechanochemical synthesis methods for similar nanocomposites is limited. The authors should expand on why this approach is advantageous compared to other techniques like hydrothermal or sol-gel methods.
- The critical analysis of existing CdS/Bi2S3 photocatalysts could be strengthened. A table comparing the performance metrics (e.g. hydrogen evolution rates, quantum efficiencies) of previously reported materials would be valuable.
- The authors should more explicitly discuss the limitations of current CdS/Bi2S3 systems and how their work addresses these challenges.
- The rationale for the chosen milling parameters (10 minutes at 400 rpm) is not explained. Some discussion of how these parameters were optimized would strengthen the methodology.
- For the photocatalytic tests, more details on the light source (e.g. spectral distribution) and hydrogen quantification method are needed to ensure reproducibility.
- The XRD analysis (Fig. 2) shows the presence of both cubic and hexagonal CdS phases. The implications of this mixed phase composition on photocatalytic activity should be discussed.
- The proposed S-scheme mechanism (Fig. 8) is interesting, but more experimental evidence (e.g. transient absorption spectroscopy) to support this model would be valuable.
- The authors should more clearly articulate how their mechanochemical approach overcomes limitations of existing synthesis methods.
- A more quantitative comparison of the photocatalytic performance to state-of-the-art materials is needed to demonstrate the advancement.
- The introduction would benefit from a clear statement of objectives and hypotheses at the end.
- The results and discussion section could be better organized by grouping related characterization techniques (e.g. structural, optical, morphological).
- Some figures, particularly the TEM images (Fig. 6), need improved resolution and labeling.
Moderate editing of English language required.
Author Response
Comments 1: The discussion of mechanochemical synthesis methods for similar nanocomposites is limited. The authors should expand on why this approach is advantageous compared to other techniques like hydrothermal or sol-gel methods. |
Response 1: Thank you for pointing this out. We agree with this comment. Therefore, we have added a paragraph about the advantages of mechanochemical method. The following text has been added to the introduction part of the manuscript: “Nevertheless, the prevailing synthesis techniques, including sol-gel, hydrothermal, and chemical bath deposition, are encumbered by considerable limitations [37]. These include the utilization of toxic solvents, intricate reaction conditions, and scalability challenges. In contrast, mechanochemical synthesis offers a solvent-free, environmentally friendly, and scalable alternative [38,39]. By employing mechanical energy to propel the reaction, our methodology obviates the necessity for deleterious solvents, streamlines the process, and permits synthesis under ambient conditions [36]. Moreover, the ball milling process fosters superior contact between CdS and Bi2S3, thereby enhancing charge separation and augmenting photocatalytic efficacy”. |
Comments 2: The critical analysis of existing CdS/Bi2S3 photocatalysts could be strengthened. A table comparing the performance metrics (e.g. hydrogen evolution rates, quantum efficiencies) of previously reported materials would be valuable. |
Response 2: Thank you for pointing this out. We agree with this comment. Therefore, we performed a literature review on the photocatalytic hydrogen ability of all CdS/Bi2S3 nanocomposites and compared their experimental results with others in the review-table (table 1). The following text has been added to the text of the manuscript: “Table 1 offers a detailed comparative review of the photocatalytic hydrogen generation rates observed in the present study, in comparison with those reported in the literature for several CdS, Bi2S3 nanoparticle, and CdS/Bi2S3 nanocomposite synthesis methods. The collected data suggests that the photocatalytic hydrogen evolution capability of the sample obtained in the present study is on par with that achieved through alternative methods. Moreover, the eco-friendly nature of this approach, which avoids the use of hazardous organic solvents, positions it as a more favorable option compared to some of the methods presented in the table. ”. |
Comments 3: The authors should more explicitly discuss the limitations of current CdS/Bi2S3 systems and how their work addresses these challenges. |
Response 3: The authors thank the Reviewer for the valuable comments. A short discussion about the limitations of CdS/Bi2S3 system was added to the introduction part of manuscript. The following text has been added to the introduction part of the manuscript: “However, CdS/Bi₂S₃ system has notable limitations, including photocorrosion, poor stability, rapid charge recombination, and inefficient charge transfer. Other challenges involve scalability issues, limited light absorption, Bi₂S₃ degradation, and suboptimal photoelectrochemical performance, all hindering practical applicability [23]”. To address one of the issues, we decorated our CdS/Bi2S3 catalyst with co-catalyst Pt for improving charge separation. |
Comments 4: The rationale for the chosen milling parameters (10 minutes at 400 rpm) is not explained. Some discussion of how these parameters were optimized would strengthen the methodology. |
Response 4: Thank you for pointing this out. We agree with this comment. In the methodology, we referenced our previous work, where we optimized the experimental parameters by adjusting the synthesis time, ball-to-powder ratio, and rotation speed of milling chamber (rpm) for the CdS nanoparticle synthesis. The following text has been added to the methodology part of the manuscript: “These parameters were found to be optimum in previous work [38] by adjusting the synthesis time, ball-to-powder ratio, and rotation speeds of milling chamber to achieve effective size reduction while minimizing particle agglomeration and heat generation”. |
Comments 5: For the photocatalytic tests, more details on the light source (e.g. spectral distribution) and hydrogen quantification method are needed to ensure reproducibility. |
Response 5: Thank you for your valuable comment. We have supplemented the experimental section '2.5. Photocatalytic Hydrogen Evolution Test' with more comprehensive data and detailed descriptions. Specifically, we have included quantitative information regarding the experimental setup, along with precise calculations of the photocatalytic hydrogen yield. This additional data covers the parameters used in the hydrogen production process, such as light intensity, reaction time, and catalyst mass, which allows for a more accurate assessment of the photocatalytic efficiency. These improvements ensure that the methodology and results are clearly understood and reproducible, providing a solid foundation for evaluating the hydrogen evolution performance under the experimental conditions. |
Comments 6: The XRD analysis (Fig. 2) shows the presence of both cubic and hexagonal CdS phases. The implications of this mixed phase composition on photocatalytic activity should be discussed. |
Response 6: We thank the reviewer for valuable comments. We agree with this comment. Thus, the implications of this mixed phase composition on photocatalytic activity was discussed. The following text has been added to the manuscript: “The XRD analysis (Figure 2) revealed the presence of two distinct crystal structures of CdS in our samples, namely the cubic (hawleyite) and hexagonal (greenockite) phases. These polymorphs display disparate electronic properties, notably in their conduction band minimum (CBM) and valence band maximum (VBM). The hexagonal CdS phase typically exhibits a higher CBM and a more negative VBM in comparison to the cubic phase [64]. Although these discrepancies are relatively minor, typically within a few tenths of an electronvolt, they can impact photocatalytic behaviour, particularly with regard to band gap and charge separation efficiency [65]. In the context of the CdS/Bi₂S₃ composite, both CdS phases contribute to the overall photocatalytic activity. The coupling with Bi₂S₃ is anticipated to facilitate charge transfer, thereby mitigating any potential adverse effects resulting from the differences in band positions between the two CdS polymorphs. Furthermore, the enhanced stability of hexagonal CdS, as indicated by the Rietveld refinement results, suggests its favorable role in forming heterojunctions, which can improve charge separation efficiency and, consequently, the overall photocatalytic performance of the composite system [66,67]. ”. |
Comments 7: The proposed S-scheme mechanism (Fig. 8) is interesting, but more experimental evidence (e.g. transient absorption spectroscopy) to support this model would be valuable. |
Response 7: Thank you for your comment! Due to time constraints at our university laboratory, we were unable to perform transient absorption spectroscopy analysis. However, we conducted XPS analyses on the CdS and Bi2S3 samples, including XPS valence band spectra, the maximum value of the tangent intersection, and the minimum value of the tangent intersection for both samples. The results are presented in Figure 5, and the corresponding discussion of these findings is provided above the figure. These results also influenced the construction of the energy diagram, Figure 9 was edited. In the mechanism discussion, we included additional text where we indirectly demonstrate the dominance of the S scheme in the hydrogen generation process. The following text has been added to the manuscript: “The conduction band energy of bismuth sulfide (Bi2S3) is below zero in the energy diagram, making hydrogen reduction on the surface of bismuth sulfide nanoparticles, when used as a catalyst, unfeasible. This conclusion is supported by the absence of detectable hydrogen in photocatalytic experiments involving bismuth sulfide with platinum deposition (Bi2S3/Pt), Figure -10. As a result, the dominant mechanism for hydrogen generation is attributed to the S-scheme. In this mechanism, during photocatalytic hydrogen generation using the (CdS/Bi2S3)Pt composite, hydrogen reduction primarily occurs at the surface of cadmium sulfide (CdS) nanoparticles. This facilitates the efficient transfer of excited electrons from CdS to platinum, enabling hydrogen reduction to proceed effectively at the Pt sites”. |
Comments 8: The authors should more clearly articulate how their mechanochemical approach overcomes limitations of existing synthesis methods. |
Response 8: Thank you for pointing this out. We agree with this comment. Therefore, we have added a text about the advantages of mechanochemical method over existing synthesis methods. The following text has been added to the introduction part of the manuscript: “Nevertheless, the prevailing synthesis techniques, including sol-gel, hydrothermal, and chemical bath deposition, are encumbered by considerable limitations [37]. These include the utilization of toxic solvents, intricate reaction conditions, and scalability challenges. In contrast, mechanochemical synthesis offers a solvent-free, environmentally friendly, and scalable alternative [38,39]. By employing mechanical energy to propel the reaction, our methodology obviates the necessity for deleterious solvents, streamlines the process, and permits synthesis under ambient conditions [36]. Moreover, the in situ ball-milling process fosters superior contact between CdS and Bi2S3, thereby enhancing charge separation and augmenting photocatalytic efficacy”. |
Comments 9: A more quantitative comparison of the photocatalytic performance to state-of-the-art materials is needed to demonstrate the advancement. |
Response 9: Thank you for pointing this out. We agree with this comment. Therefore, we have added a discussion about quantitative comparison of the photocatalytic performance to state-of-the-art materials. The following text has been added to the introduction part of the manuscript: “The development of photocatalytic hydrogen evolution has led to the creation of a number of highly effective semiconductor materials, which have the capacity to markedly enhance the efficiency of hydrogen production. Among these, the Cu/CdS/MnOx heterostructure is particularly noteworthy, exhibiting an exceptional hydrogen evolution rate (HER) of 5965.03 μmol h⁻¹g⁻¹. This is largely attributed to its effective charge separation mechanisms, which optimize the utilization of photogenerated carriers [70]. Similarly, a ZnIn2S4 photocatalyst integrated with single-atom catalysts has demonstrated exceptional performance, achieving a HER of up to 12,416.8 μmol h-1g-1 and an AQE of 4.9%. This performance highlights the significant advantages of single-atom activation in enhancing photocatalytic activity [71]. Furthermore, the W18O49/S heterojunction has been documented to demonstrate robust performance with an HER of 5373 μmol h-1g-1, which serves to exemplify the advancements in photocatalytic systems [70]. In comparison, the CdS/Bi2S3/Pt composite displays a HER of 996.68 μmol h-1g-1 and an AQE of 0.87%. Although these findings do not rank our composite among the most effective photocatalysts, they nevertheless demonstrate its potential for further enhancement. The current performance indicates that with targeted improvements and innovative modifications, this composite could be developed into a more efficient photocatalyst in future research endeavours [72]”. |
Comments 10: The introduction would benefit from a clear statement of objectives and hypotheses at the end. |
Response 10: The authors thank the Reviewer for the valuable comments. We agree with this comment. Thus, the last paragraph of the introduction part was modified clearly indicating the aim and hypothesis of this work. The following text has been added to the introduction part of the manuscript: “This study aims to develop a cost-effective, solvent-free, and scalable method for synthesizing Bi2S3-decorated CdS nanoparticles with enhanced photocatalytic hydrogen generation capabilities. We hypothesize that the mechanochemical approach, in conjunction with thermal annealing, will improve charge separation and minimize electron-hole recombination, thereby enhancing hydrogen evolution performance”. |
Comments 11: The results and discussion section could be better organized by grouping related characterization techniques (e.g. structural, optical, morphological). |
Response 11: The authors thank the Reviewer for the comment. The current structure was selected to ensure a logical flow that aligns with the experimental progression and narrative of the study. While grouping methods (e.g., structural, optical, morphological) is a valid approach, we believe that the current structure maintains consistency between results and analysis, thereby increasing the clarity of our interpretation. Therefore, we respectfully propose that the current structure should be retained. |
Comments 12: Some figures, particularly the TEM images (Fig. 6), need improved resolution and labeling. |
Response 12: The authors thank the Reviewer for the comments. We agree with this comment. Despite our best efforts, we were unable to enhance the resolution of the TEM images in Figure 6 due to limitations in the original data. However, we have focused on improving the labelling. The figure has been updated with clearer and more comprehensive labels to better guide the reader's understanding of the observed structures. |

Round 2
Reviewer 2 Report
Comments and Suggestions for Authors
Accept
Author Response
Comments 1: Figure 1: "1 step" and "2 step" should be corrected to "1st step" and "2nd step" respectively. |
Response 1: The authors thank the Reviewer for the comment. We agree with this comment. Therefore, in Figure 1, "1 step" and "2 step" was corrected to "1st step" and "2nd step". |
Comments 2: Some important information is missing in some references: 30,34,55,71 |
Response 2: The authors thank the Reviewer for the comment. We agree with this comment. All references were corrected. The following changes were made to upgrade references: “34. Zhang, N.; Lü, J.; Cao, R. Photocatalytic CO2–to–ethylene conversion over Bi2S3/CdS heterostructures constructed via facile cation exchange. Research 2022, 2022. 38. Uddin, M.R.; Khan, M.R.; Rahman, M.W.; Cheng, C.K.; Yousuf, A. Photocatalytic conversion of CO2 into methanol: Significant enhancement of the methanol yield over Bi2S3/CdS photocatalyst. International Journal of Engineering Technololgy and Sciences 2015, 3. 59. Zuo, X.-Q.; Yang, X.; Zhou, L.; Yang, B.; Li, G.; Tang, H.-B.; Zhang, H.-J.; Wu, M.-Z.; Ma, Y.-Q.; Jin, S.-W. Facile synthesis of Bi₂S₃–C composite microspheres as low-cost counter electrodes for dye-sensitized solar cells. RSC Advances 2014, 4, 57412-57418. 75. Cui, J.; Wang, Y.; Lin, L.; Yang, X.; Luo, X.; Guo, S.; Xu, X. Single-atomic activation on ZnIn2S4 basal planes boosts photocatalytic hydrogen evolution. Nano Research 2024, 17, 5949-5955.” |
Comments 3: Some closely related important publications are missing, such as: a) Dual interfacial electric fields in black phosphorus/Ti3C2 MXene/MoB MBene ultrathin ternary composite enhances broad-spectrum carrier migration efficiency of photocatalytic devices, Device 2024, 1, 100283. |
Response 3: The authors thank the Reviewer for the comment. The following text was modified in the Introduction part of the manuscript: “These nanomaterials, such as TiO2, ZnO, g-C3N4, and CdS, and their binary [13,14] and ternary composites [15,16] are designed to absorb sunlight and facilitate the separation of water molecules into hydrogen and oxygen [17].” |
